# Current Limitations of Sentinel Node Biopsy in Vulvar Cancer

**DOI:** 10.3390/curroncol32040215

**Published:** 2025-04-08

**Authors:** Myriam Gracia, Maria Alonso-Espías, Ignacio Zapardiel

**Affiliations:** Gynecologic Oncology Unit, La Paz University Hospital, 28015 Madrid, Spain; maespias@salud.madrid.org (M.A.-E.); ignacio.zapardiel@salud.madrid.org (I.Z.)

**Keywords:** vulvar cancer, sentinel lymph node biopsy, recurrent vulvar cancer, inguinofemoral lymphadenectomy

## Abstract

**Background:** Vulvar cancer is a rare gynecologic malignancy with increasing incidence. Lymph node status is the most critical prognostic factor, traditionally assessed through inguinofemoral lymphadenectomy, a procedure associated with significant morbidity. Sentinel lymph node biopsy (SLNB), in selected cases, has emerged as a less invasive alternative with favorable oncologic outcomes. **Objective:** This review summarizes current evidence on the indications, technique, safety, and oncologic outcomes of SLNB in vulvar cancer, with a focus on controversial scenarios such as recurrent and larger tumors. **Methods:** A narrative review of PubMed-indexed studies published in English over the last 35 years was conducted. Eligible studies included original research, systematic reviews, meta-analyses, randomized controlled trials, and case-control studies. **Results:** SLNB is recommended for unifocal vulvar tumors < 4 cm with stromal invasion > 1 mm and clinically negative nodes. Landmark trials, including GROINSS-V-I and GOG-173, confirmed its accuracy and lower morbidity compared to lymphadenectomy. SLNB utilization has increased since its inclusion in guidelines, with a concurrent decline in lymphadenectomy rates. Combined detection techniques are mandatory, while indocyanine green (ICG) is an emerging option. Future studies should focus on refining patient selection criteria, improving detection techniques, and clarifying the implications of low-volume nodal disease to further optimize outcomes for patients with vulvar cancer. **Conclusion:** SLNB is a validated, minimally invasive staging approach in early-stage vulvar cancer. Further research is needed to refine its role in high-risk cases and optimize detection methods.

## 1. Introduction

Vulvar cancer is a relatively uncommon neoplasm, ranking 21st in incidence among women worldwide and representing only 4% of all gynecologic malignancies. According to the Global Cancer Statistics, 47,342 new cases and 18,579 deaths related to vulvar cancer were reported in 2022 [1]. Despite its rarity, its incidence appears to be increasing in recent decades, likely due to higher life expectancy and human papillomavirus infection [2].

In addition to tumor excision, inguinofemoral lymph node assessment is crucial for tumors with stromal invasion greater than 1 mm, as lymph node metastases are found in approximately 23–25% of patients with early-stage disease. The first sites of lymphatic spread in vulvar cancer are the inguinofemoral lymph nodes, followed by the pelvic nodes. The risk of unilateral or bilateral nodal involvement is influenced by factors such as tumor size and proximity to the midline or clitoris [3]. Furthermore, lymph node involvement is the most significant independent prognostic factor, with survival rates decreasing from 90% to less than 60% in cases of groin metastases [4]. Key factors strongly associated with nodal involvement include lymphovascular space invasion, tumor stage, tumor grade, and depth of infiltration [5,6].

Traditionally, the standard treatment for vulvar cancer consists of a radical vulvectomy combined with bilateral inguinofemoral lymphadenectomy. This procedure involves the removal of superficial and deep inguinal lymph nodes located within the femoral triangle, which is bordered by the inguinal ligament, sartorius, and adductor longus muscles. However, this approach is linked to a significant increase in both short- and long-term morbidity, including wound dehiscence, infections, lower limb lymphedema, and lymphocele formation, all of which negatively impact patients’ quality of life [7,8].

Sentinel lymph node biopsy (SLNB) has emerged in recent decades as a less invasive alternative for nodal assessment in various malignancies, including melanoma, breast, and endometrial carcinoma, but also in vulvar carcinoma, reducing morbidity associated with lymphadenectomy [9,10,11]. Decesare et al. were the first to describe sentinel lymph node identification using lymphoscintigraphy in vulvar cancer in 1997 [12]. Several studies have demonstrated that sentinel lymph node biopsy (SLNB) is a safe, accurate, and cost-effective approach for managing early-stage vulvar cancer. In tumors smaller than 4 cm, SLNB has a false-negative rate of only 2% and a negative predictive value of 98% [13,14,15,16]. Furthermore, the 2008 Groningen International Study on Sentinel Nodes in Vulvar Cancer (GROINSS-V-I) showed that omitting inguinofemoral lymphadenectomy in patients with early-stage vulvar cancer who have negative sentinel lymph nodes is a safe alternative. This approach is associated with a three-year survival rate of 97% and a nodal recurrence rate of only 2.3% [13]. Additionally, it showed that avoiding lymphadenectomy significantly reduces treatment-related morbidity, including lower rates of lymphedema and wound complications.

Based on these results, major international guidelines currently recommend exclusive SLNB for unifocal vulvar tumors smaller than 4 cm with stromal invasion greater than 1 mm and no clinical suspicious nodes (palpation/imaging) [17,18]. Inguinal lymphadenectomy should be completed in cases where there is no tracer migration, or if the pathological examination reveals the presence of metastases in the sentinel lymph node (SLN). For lesions > 4 cm or multifocal tumors, bilateral lymphadenectomy remains the primary standard procedure.

For the SLN assessment, the use of a radiotracer (typically Tc99m nanocolloid) is mandatory, while combination techniques with isosulfan or methylene blue dye or indocyanine green (ICG) are recommended, as they have been associated with improved accuracy [19,20].

In multifocal tumors or tumors larger than 4 cm, however, the use of SLNB remains a subject of debate due to the limited available evidence regarding its oncologic safety in these scenarios [17]. Increased tumor size is associated with a higher risk of lymph node metastasis, while multifocality may lead to multiple lymphatic drainage pathways, both of which could potentially compromise the diagnostic accuracy of the technique [21]. Similarly, there are no established consensus statements regarding the use of inguinofemoral SLNB in patients with prior vulvar excision. As a result, lymphadenectomy remains the standard treatment for patients with recurrent vulvar squamous cell carcinoma.

This review aims to summarize the latest evidence on the indications, technique, safety, and oncologic outcomes of SLNB in vulvar cancer, with a particular focus on controversial scenarios such as recurrent disease and larger or multicentric tumors.

## 2. Materials and Methods

A comprehensive review of the literature was performed using the PubMed database, using terms such as “vulvar cancer”, “sentinel node biopsy”, “inguinofemoral lymphadenectomy”, and “recurrent vulvar cancer” in different combinations. The search has been limited to original studies, systematic reviews, meta-analyses, randomized controlled trials, and case controls published in English within the last 35 years. Publications relevant to the search and their cited references were retrieved and evaluated independently by M.G. and M.A. for inclusion in the text.

## 3. Results

### 3.1. Current Indications and Evidence for Sentinel Node Biopsy

SLNB is a minimally invasive surgical technique applicable to women with early-stage vulvar cancer. The European Society of Gynecologic Oncology (ESGO) and the National Comprehensive Cancer Network (NCCN) guidelines currently recommend SLNB for patients with unifocal vulvar tumors smaller than 4 cm and stromal invasion greater than 1 mm, with clinically and radiologically negative groins [17,18]. In patients with small lateralized tumors (<4 cm in size and ≥1 cm from the vulvar midline) and negative ipsilateral lymph nodes, the risk of contralateral groin node metastasis is below 1% [3]. Therefore, ipsilateral groin dissection is considered sufficient in these cases. Conversely, for non-lateralized tumors, a bilateral surgical evaluation is recommended. The intraoperative frozen section of the SLN is optional. The risk–benefit balance between accurately assessing the size of lymph node metastases through frozen section analysis and the potential tissue loss, along with the possible need for a second surgical intervention if the SLN is positive, should be carefully evaluated. In case the SLN is not detected, inguinofemoral lymphadenectomy, including superficial and deep nodes, should be performed. Similarly, if detection is achieved on only one side in midline tumors, contralateral lymphadenectomy is mandatory [17,18].

During the 2000s, several studies provided promising data on the detection rate and false-negative rates of SLNB in patients with vulvar cancer. However, the majority were retrospective studies with a limited sample size [22,23,24,25,26].

It was not until 2008 that van der Zee et al. published the results of the GROINSS-V-I study, the largest validation multicenter observational study that investigated the safety and utility of SLNB in early-stage vulvar cancer [13]. This study included 403 patients with <4 cm squamous cell carcinoma of the vulva and clinically nonsuspicious inguinofemoral lymph nodes. In 259 of these patients, the SLN was negative after pathological ultrastaging, and no further lymphadenectomy was performed. The 3-year survival rate in this group was 97%, with only a 2.3% rate of nodal recurrence after a median follow-up of 35 months. In addition, both short- and long-term morbidity was significantly reduced when inguinofemoral lymphadenectomy was omitted (lymphedema 1.9% vs. 25.2%, and recurrent erysipelas 0.4% vs. 16.2%), demonstrating that SLNB was safe and less harmful compared with inguinofemoral lymphadenectomy in unifocal, <4 cm vulvar carcinomas. Furthermore, in the long-term follow-up of the GROINSS-V-I trial, Grootenhuis et al. reported 5-year and 10-year disease-specific survival rates of 93.5% and 90.8%, respectively, for patients with negative SLN, confirming the long-term safety of SLNB in this population. Despite the positive survival outcomes, the study emphasized that patients still faced a risk of recurrence. They reported a local recurrence rate of 24.6% and 36.4% at 5 and 10 years, respectively, for SLN-negative patients. This underscores the importance of long-term monitoring, even more than a decade after the initial treatment [27].

Another key study that contributed to establishing SLNB as the standard technique for lymph node staging in vulvar cancer was the Gynecologic Oncology Group (GOG-173) trial, whose findings were published in 2012 [14]. This study included 452 women with squamous cell carcinoma of the vulva, with tumor sizes ranging from 2 to 6 cm. All participants underwent SLNB followed by inguinal lymphadenectomy. The sensitivity of the technique for detecting metastases was 92.5% overall, with a false-negative rate of 3.7%, which decreased to 2% in tumors smaller than 4 cm, thus corroborating the results of the GROINSS-V-I study.

More recently, additional studies have demonstrated the safety of SLNB as an exclusive method for lymph node staging in cases with negative results. A systematic review and meta-analysis conducted in 2015 by Covens et al., which included 11 studies, found that the rate of groin recurrence was not significantly higher for SLNB (3.4%) compared to complete inguinofemoral lymphadenectomy (1.4%) [28]. Another systematic review and meta-analysis, incorporating 29 studies with 1779 patients, reported an overall SLNB sensitivity of 95% (95% CI: 92–98%). However, they observed a 9% false-negative rate, emphasizing that this relatively high rate underscores the impact of the learning curve, a persistent challenge in managing rare cancers [15]. Table 1 shows the main studies evaluating the safety of SLBN in vulvar cancer.

The inclusion of SLNB as a method for lymph node staging in vulvar cancer was added to international guidelines in 2016 [30]. Since then, the adoption of this technique has progressively increased, as demonstrated by a recent study that analyzed the implementation of surgical de-escalation in gynecologic cancers using the National Cancer Database. They reported that the utilization rate of SLNB in vulvar cancer rose from 12.3% to 36.9% between 2012 and 2020, while lymphadenectomy rates simultaneously declined from 87.7% to 63.2% [31]. These findings reflect a growing shift towards less invasive approaches in the management of early-stage vulvar cancer, aiming to minimize surgical morbidity without compromising oncological outcomes. Furthermore, with the reduction in morbidity, SLNB has also been shown to be more cost-effective compared to lymphadenectomy [32,33].

### 3.2. Surgical Technique of Sentinel Node Biopsy and Available Methods of Detection

The most accurate detection methods for SLNB in appropriately selected patients with vulvar cancer are combined techniques [17].

Current evidence supports the use of a combination of blue dye and 99m-Tc nanocolloid for sentinel lymph node detection, in accordance with the methodology recommended by the GROINSS-V study [13] and international guidelines [17,34]. The standard protocol involves injecting 4 mL of blue dye and 4 mL of 2.5 mCi Technetium-99, divided into four 1-mL intradermal aliquots placed in four orthogonal peritumoral locations. To ensure adequate migration to the lymph nodes, Technetium-99 should be administered preoperatively, while blue dye is injected just before groin dissection during surgery under anesthesia [34]. The sentinel lymph node must be removed before the primary tumor is excised. A groin incision is performed at the site with the highest signal intensity detected by gamma probes, followed by careful dissection to trace lymphatic pathways and locate the sentinel node. The gamma detector should then confirm the presence of radioactivity in the excised node and verify that no residual tracer remains in the lymphatic basin. If multiple sentinel nodes are detected, all should be removed [34,35]. A node detected as “positive” by the gamma probe is defined as having a count at least five times greater than the background level. After removing the targeted node, the background reading with the gamma probe should be less than 10% of the node’s count. While locating the node(s), it is crucial to consider its orientation toward the vulva to avoid mistaking background vulvar uptake for the node itself [36]. With the growing adoption of inguinofemoral SLN detection, the techniques used for identification have also advanced. Initially, the most commonly employed methods included blue dyes, such as lymphazurin and methylene blue, along with radiocolloid lymphoscintigraphy [27]. However, since the introduction of near-infrared imaging with indocyanine green (ICG) in 2010, this approach has gained widespread acceptance for SLN detection in the inguinofemoral region [15]. Regardless of the specific mapping technique, SLNB in vulvar cancer generally demonstrates high detection and mapping success rates. The selection of the preferred method is primarily influenced by the surgeon’s expertise, training, and institutional experience [37]. As previously noted, growing evidence supports the use of ICG as a viable alternative to blue dye. Next, we are going to summarize some of the current evidence on the different tracers for SLNB in vulvar cancer. In 2010, Crane et al. introduced the use of an infrared light source and camera for intraoperative detection of inguinofemoral SLNs labeled with ICG (Figure 1). They later published their findings on SLN mapping in vulvar cancer, analyzing 16 groins using a combination of 99m-Tc, blue dye, and ICG with near-infrared imaging. Their study identified 29 SLNs with 99m-Tc, 26 with ICG, and 21 with blue dye [38,39]. Following this, in 2017, Soergel et al. reported their experience with ICG in a cohort of 27 patients. A key finding of their study was the identification of eight SLNs that were not detected by 99m-Tc but were successfully visualized using ICG alone [40].

Recently, Deken et al. demonstrated, in their randomized trial, comparable efficacy in terms of sentinel node detection between isotope/ICG compared with conventional detection method with isotope/blue dye. They randomized 48 patients to the blue dye (n = 24) or ICG group (n = 24), using in both groups also 99m-Tc nanocolloid. The SLN detection rate was 92.1% of the groins in the blue dye group and 97.2% in the ICG group (*p* = 0.33). Surgical outcomes were similar between groups, although more short-term postoperative complications (wound infection or breakdown and lymphocyst formation) were observed in the blue dye group (*p* = 0.041) [19].

A systematic review of the literature aimed to determine if the use of ICG alone in detecting SLN in vulvar cancer was as accurate as the gold standard dual-labeling technique; 13 studies were identified with similar detection rates for SLN to the gold standard technique (ranged between 89.7 and 100% among the different studies). The authors also highlighted the potential of decreasing the detection rate in two settings; in the most metastatic lymph nodes and in obese patients and midline tumors. No adverse events were reported, and no consensus regarding to ICG injection site or timing, volume or concentrations, or use of serum albumin or hybrid tracer was reached [41]. A recent meta-analysis by Di Donna et al. provides additional evidence supporting the effectiveness of ICG in SLN detection for vulvar cancer. The study aimed to identify the technique with the highest detection rate by comparing planar lymphoscintigraphy (PL), blue dye, and ICG fluorescence. After selecting 30 studies, the researchers found that the SLN detection rates per patient and per groin were 96.13% and 92.57% for PL, 90.44% and 66.21% for blue dye, and 91.90% and 94.80% for ICG, respectively. While the patient-based analysis did not reveal significant differences between these methods, the groin-based analysis showed that both PL and ICG had significantly higher detection rates compared to blue dye (*p* < 0.05) [20]. Additionally, ICG appears to have a steeper learning curve, with detection rates improving as surgical teams gain more experience and training [42]. Moreover, a hybrid technique combining ICG with 99m-Tc nanocolloid has been introduced, allowing injection up to 20 h before surgery while still enabling intraoperative near-infrared detection [43]. Another recent approach, described in the SARVU study, involves the use of ferromagnetic tracers, which demonstrated a detection efficacy comparable to that of radiotracers [44].

The method of histopathological evaluation could influence the reliability of SLN in staging vulvar cancer. Postoperative pathological analysis of the SLN can be complemented with intraoperative frozen section assessment. If the SLNB is positive, this technique provides the advantage of allowing an immediate complete lymphadenectomy. However, it has a relatively high false-negative rate and poses a risk of losing diagnostic tissue. A precise pathological evaluation of SLNs is essential. As observed in GOG 173 [14] and GROINS V [13], hematoxylin and eosin staining of lymph nodes fails to detect at least 20% of nodal metastases compared to additional immunohistochemical (IHC) analysis. Further serial sectioning could identify an extra 7% of nodal metastases [45]. Ultrastaging of lymph nodes, which incorporates serial sectioning and immunohistochemistry, is an important element of the SLN technique. A review demonstrated that the detection rate of micrometastases is significantly higher when combining hematoxylin and eosin staining with serial sectioning and immunohistochemistry [46]. Thus, ultrastaging of the nodes, with serial sectioning and staining, is recommended for the identification of possible micrometastases and isolated tumor cells [17]. The clinical relevance of micrometastases identified through ultrastaging remains a topic of debate. Isolated groin recurrence has been documented in only a few cases involving patients with micrometastases in the SLN [46].

Although the findings from this retrospective series regarding near-infrared imaging performance are encouraging, further prospective studies are necessary to assess the effectiveness of indocyanine green and near-infrared imaging, thereby defining the role of SLN mapping in this condition. Protocols for using ICG are still varied, and the ideal approach has yet to be established. Currently, the combination of an isotope with either blue dye or ICG appears to achieve the highest detection rates and demonstrates proven clinical effectiveness. Performing pre-operative lymphoscintigraphy is advised to determine the number and location of sentinel nodes.

### 3.3. Management of Metastatic Groin Nodes

Additional treatment of nodes in the groin is determined based on the outcomes of the SLNB. In the case of negative SLN, no additional or adjuvant treatments are necessary.

#### 3.3.1. Volume of Lymph Node Disease

The presence of lymph node metastasis is associated with poorer disease-free survival and overall survival rates. Metastatic lymph node disease in early-stage vulvar cancer is diagnosed in 10.7% of stage I patients and 26.2% of stage II patients, affecting between 21% and 35.8% of patients [47]. According to Mahner et al., the disease-free survival rate is 35.2%, and overall survival is 56.2%, compared to 75.2% and 90.2% in patients without lymph node involvement [48]. The lymph node metastases are classified as stage III disease. The new FIGO [3] classification considers the presence and count of metastatic lymph nodes (≤2 and >2 lymph nodes) and the volume of lymph node disease (<5 and ≥5 mm). This is based on the results of the GROINS-V trial [49]. The risk of metastatic affectation of a different node than the sentinel one increases not only with the number of metastatic SLN but also with the volume of the disease. Moreover, the authors found statistically significant differences in the disease-specific survival after 24 months of follow-up between patients with SLN metastases ≤ 2 and >2 mm (69.5% vs. 94.4%, respectively).

The management of metastatic lymph nodes is controversial; however, additional treatment (complete inguinofemoral lymphadenectomy or adjuvant treatment) to the corresponding inguinofemoral area should be promptly administered after finding metastatic disease in an SLN in patients with vulvar carcinoma [17,20]. The role of adjuvant radiotherapy was evaluated in a large multicenter retrospective study. An improvement in prognosis, in both disease-free survival (DFS) and overall survival (OS), was observed in patients with positive lymph nodes who underwent postoperative radiotherapy [42].

Patients with macrometastasis (>2 mm) or extracapsular spread typically require completion inguinofemoral lymphadenectomy, followed by adjuvant radiotherapy if more than one node with macrometastasis and/or extranodal spread is found in the final pathological report. However, recent studies suggest that radiotherapy alone may be a viable option for selected patients with micrometastatic disease (≤2 mm) to reduce surgical morbidity without compromising oncologic outcomes. Recently, the large prospective multicenter GROINSS-V II study aimed to establish the safety of replacing inguinofemoral lymphadenectomy with radiotherapy for patients with early-stage vulvar cancer with metastasis in an SLN. An analysis of isolated groin recurrence in the first 91 patients found that nine out of ten had macrometastatic disease (>2 mm). As a result, the protocol was revised to restrict inguinofemoral node radiotherapy without further surgery (complete inguinofemoral lymphadenectomy) only to patients with micrometastatic disease (≤2 mm) in the SLN. Patients with micrometastases were treated with inguinofemoral lymphadenectomy as the standard approach, with additional radiotherapy for those who had multiple node metastases or extracapsular spread. The authors report a median follow-up period of 24.3 months. Among patients with SLN micrometastases (≤2 mm) who received radiotherapy alone without lymphadenectomy, recurrence rates were low (1.6%), with manageable levels of treatment-related toxicity. Among 162 patients with SLN macrometastases, the isolated groin recurrence rate at 2 years was 22% in those who underwent radiotherapy, and 6.9% in those who underwent IFL (*p* = 0.01) [50].

Currently, the GROINSS-V III study is examining whether patients with an SLN macrometastasis can be managed with chemoradiation instead of radiotherapy to both prevent groin recurrences and reduce treatment-related morbidity [51]. The NCCN already advises the use of concurrent chemotherapy with radiotherapy for treating SLN metastases. However, this recommendation is primarily based on studies involving patients with advanced vulvar cancer. So far, chemotherapy and chemoradiation have not been sufficiently researched as treatment options for SLNmetastases in early-stage vulvar cancer [17,18].

#### 3.3.2. Management of Contralateral Groin

The current evidence on the safety of omitting treatment to the no metastatic groin in patients with unilateral positive groin after bilateral SLNB is also controversial and conflicting. Current guidelines recommend that in cases of unilaterally positive SLN, bilateral inguinofemoral lymphadenectomy should be performed, increasing the risk of associated morbidity [17,18]. The percentage of contralateral non-sentinel node affectation varies among the different studies. In these cases, the risk of contralateral lymph node affectation seems to be low according to some authors. The rate of contralateral groin positivity in the studies carried out by Woelber, Nica, and Ignatov ranges between 0 and 5.3%. These data support the omission of contralateral inguinofemoral lymphadenectomy to reduce surgical morbidity and long-term complications [52,53,54]. In contrast, data from a German study identified contralateral positivity in a higher percentage of patients with midline vulvar carcinoma—a total of 4 patients out of 18 (22%). Moreover, they also found that the depth of tumor infiltration correlated significantly and positively with the incidence rate of contralateral groin metastasis (*p* = 0.0038) [55].

Recently, a large prospective study from the GROINSS-V group provides evidence to omit treatment to the unaffected contralateral groin, in cases where bilateral drainage has been mandatorily identified for midline tumors. In this study, the contralateral non-SLN metastases/groin recurrence rate was 2.9%. Therefore, they conclude that omitting contralateral lymphadenectomy in these cases should be safe, although caution is advised with tumors of >3 cm, the authors highlighting that the majority of non-sentinel contralateral metastasis occurred in these larger tumors [21].

### 3.4. Extended Indications for Sentinel Node Biopsy

Although SLN biopsy is currently recommended for unifocal tumors under 4 cm, research is exploring its feasibility in patients with larger or multifocal tumors.

#### 3.4.1. Sentinel Node Biopsy in Recurrent Vulvar Cancer

Some studies have also suggested that SLNB may still be a reliable staging method in carefully selected cases of the first recurrence of vulvar cancer, potentially reducing the need for full lymphadenectomy in a broader patient population. However, further validation through prospective trials is required before guideline expansion. At present, evidence for the use of the SLN procedure in the case of recurrent cancer is lacking. A small retrospective study suggests that the technique is feasible, but that detection rates are lower and lymphatic drainage may be unusual after surgery of the primary tumor [17,56]. Inguinofemoral lymphadenectomy is considered the standard treatment for patients with recurrent vulvar cancer who did not undergo this procedure during the first episode. Many of the patients with vulvar cancer (primary or relapse episode) are elderly and frail patients; for these reasons, an alternative treatment to lymphadenectomy should be proposed. However, at the moment, the evidence about the accuracy and safety of repeat SLNB in recurrent disease is still lacking. The main reason not to perform a repeat SLNB in patients with a local recurrence of vulvar cancer is the assumption that the lymph flow might be altered because of previous surgery or radiotherapy [56]. Otherwise, SLNB, also in recurrent disease, could add some advantages such as improving the visualization of the lymph drainage and guiding the surgeon in the removal of the lymph nodes at risk. Until now, there are no prospective studies assessing the efficacy and safety of SLNB in recurrent vulvar cancer, and only two case reports and a small retrospective study have been published regarding this topic [23,56,57]. In 2016, van Doorn et al. published a retrospective series of 27 patients with a first recurrence of squamous cell carcinoma of the vulva who underwent SLNB. They found this procedure seemed feasible, in that in 77% of patients and in 84% of the groins, the SLN procedure could be performed as planned. This percentage was less compared to the rates of success procedure in primary SLNB, which is around 95% [56]. The authors also observed that the procedure was technically more challenging in recurrent disease compared to initial surgery. Regarding oncological outcomes, data are lacking, but so far none of the patients in the study suffered local or distant recurrences after repeating SLNB during a follow-up of a median of 27 months (range 2–96 months).

Currently, there is an ongoing trial designed to prove the feasibility and safety of SLNB in recurrent vulvar cancer. The outcomes of this current prospective study will help to bridge knowledge gaps and define future research questions. If the SLN procedure is confirmed to be safe and feasible in this patient group, it will significantly help reduce both the short- and long-term side effects of vulvar cancer treatment while having a smaller impact on quality of life compared to the current standard treatment. Additionally, we expect the study to enhance our understanding of the efficacy, side effects, and pathology of recurrent vulvar cancer [58].

#### 3.4.2. Sentinel Node Biopsy in >4 cm or Multifocal Tumors

The safety of the SLN procedure in squamous cell carcinoma of the vulva > 4 cm or multifocal tumors is also lacking and at the moment the recommendation is to perform inguinofemoral lymphadenectomy. This evidence is based on the study GOG-173 where the authors found a detection rate of 92.0% and seven false-negative SLNs, resulting in an unacceptable false-negative predictive value of 7.4% in patients with tumors between 4–6 cm of diameter [14]. Several prospective studies investigating SLN biopsy in vulvar cancer have reported detection rates varying between 88% and 97% per patient and have calculated a pooled detection rate of 93% per groin [13,14,20,59]. Recently, a prospective pilot study has been carried out to assess if indications for SLNB in vulvar cancer could be extended [60]. They found detection rates and negative predictive values of SLN biopsy for women with primary squamous cell vulvar cancer ≥ 4 cm and multifocal tumors comparable to the results for smaller, unifocal tumors. The detection rates varied between 100% and 94.1% per patient and between 84.1% and 85.3% per groin. Moreover, they found no false-negative SLN in the population study [60]. Another prospective study published in 2017 by Garganese et al. reported on 12 patients with tumors > 4 cm undergoing an SLN biopsy followed by inguinofemoral lymphadenectomy. In this cohort, no patients had a false negative SLN [59].

Multifocality of vulvar cancer is a rare condition which is reflected by the low number of patients collected in a few studies. Because of unsatisfactory oncological safety, the GROINSS-V-I study stopped the inclusion of multifocal tumors after two early inguinal recurrences between the hitherto 19 included patients [13]. However, Garganese et al. included nine women with multifocal tumors in their cohort, with no false-negative SLN [59].

Nowadays, due to the low evidence we have, the safety of the SLNB in squamous cell carcinoma of the vulva larger than 4 cm or in multifocal tumors is not yet fully established. Although recent studies have shown detection rates and negative predictive values comparable to those of smaller, unifocal tumors, the current recommendation remains to perform inguinofemoral lymphadenectomy.

### 3.5. Potential Barriers and Limitations for Sentinel Node Biopsy

The current standard technique for SLNB could present some cumbersome aspects. For example, the preoperative lymphoscintigraphy along with intraoperative radiolocalization requires multiple procedures that often need to be performed the day before the surgical procedure. Additionally, intraoperative detection with 99m-Tc is complicated and relies on aural cues, necessitating a pause in the procedure at frequent intervals so that 99m-Tc uptake can be measured [27]. The use of dyes facilitates the procedure. While blue dye aids in visually identifying the SLN, its effectiveness relies on the clear detection of both lymphatic vessels and the node itself. In contrast, indocyanine green combined with near-infrared imaging overcomes these limitations by offering real-time visual guidance throughout the procedure [27]. Despite its benefits, SLN biopsy has certain limitations. Factors such as disrupted lymphatic drainage from prior surgeries, the presence of large tumors, or the learning curve associated with the technique can affect its accuracy. Additionally, the limited availability of nuclear medicine facilities and experienced surgical teams poses a challenge in resource-constrained settings. Overcoming these obstacles is crucial for expanding the use of SLN biopsy and ensuring equitable access in vulvar cancer treatment.

While intraoperative frozen section analysis may be considered, particularly to prevent the need for a second surgical procedure, caution is necessary due to the risk of overlooking micrometastases in the final histological assessment and the importance of precisely measuring metastatic deposits [61].

Another important aspect of SLNB in vulvar cancer is the one related to cost-effectiveness. There are studies that demonstrated that SLNB is more cost-effective than inguinofemoral lymphadenectomy. Erickson et al. and McCann et al. found that SLNB had both a lower annual cost and lower cost/effectiveness ratio than lymphadenectomy [32,33]. This was further demonstrated when a significant reduction in complications such as lymphedema and infection was observed after performing SLNB and, in consequence, a higher score in quality-of-life questionnaires.

### 3.6. Future Directions

The morbidity of open inguinal incisions has prompted the search for a minimally invasive approach to lymph node dissection. Videoendoscopic lymphadenectomy may be a valid alternative to the open route with no differences in terms of surgical outcomes, except for operative time, which is shorter for the open approach, and wound complications, which are less frequent for the videoendoscopic route [62]. The development of a robot-assisted platform, the da Vinci Firefly, allows for intraoperative evaluation of the lymphatic channels and SLN mapping after injection of the primary tumor with ICG. A less invasive dissection of the inguinofemoral region using robot-assisted, near-infrared fluorescence; SLN mapping; and inguinal lymph node dissection may be a feasible approach to reduce short- and long-term morbidity [63,64]. Future analysis of randomized controlled trials in this specific topic and patient population should be carried out to confirm these results.

## 4. Discussion

The SLNB has emerged as a valuable technique in the management of early-stage vulvar cancer, offering a less invasive alternative to full lymphadenectomy. This approach aims to reduce morbidity while maintaining oncologic safety. The present review highlights the current evidence regarding the accuracy, benefits, and limitations of SLN biopsy in vulvar cancer.

Several studies have demonstrated that SLN biopsy provides reliable staging information with high sensitivity and specificity [5,13,14]. The GROINSS-V trials, among others, have shown that SLN biopsy is an effective method for identifying nodal metastases in early-stage vulvar cancer, significantly decreasing complications such as lymphedema and wound infections compared to complete lymphadenectomy [49,50]. These findings support the role of SLN biopsy as the standard approach in appropriately selected patients with unifocal tumors and clinically negative nodes.

SLNB in early-stage vulvar cancer is an effective and safe way to avoid significant morbidity surrounding surgical intervention without compromising oncologic outcomes. Best practices include using a dual detection technique with a radiocolloid tracer and a visual dye, which has been proposed to enhance detection and minimize false-negative rates. Surgeon comfort and experience using these techniques are imperative to ensure the safety and efficacy of this procedure. Given this, future frontiers need to include the expansion of these services through training and access to high-volume centers [36]. More recently, ICG fluorescence imaging has emerged as an alternative technique, with studies suggesting comparable detection rates to the standard dual-tracer approach.

An ongoing area of research is the long-term oncologic outcomes of patients undergoing SLN biopsy compared to those undergoing complete lymphadenectomy. The GROINSS-V II trial provided valuable insights into the management of SLN-positive patients, suggesting that radiotherapy alone may be a viable alternative to lymphadenectomy in cases of micrometastatic disease (≤2 mm) [50]. However, the management of macrometastatic disease (>2 mm) remains controversial, with ongoing research evaluating whether chemoradiation could replace lymphadenectomy in these patients (GROINSS-V III trial) [51].

Another unresolved issue is the management of the contralateral groin in patients with unilateral SLN metastases. Current guidelines recommend bilateral inguinofemoral lymphadenectomy in such cases, but emerging data suggest that omitting contralateral treatment may be safe in select patients, particularly when contralateral SLNs are negative. Prospective studies, including those from the GROINSS-V group, continue to investigate this question to optimize the balance between oncologic safety and surgical morbidity.

Despite its advantages, SLNB has limitations. False-negative results remain a concern, particularly in cases with large tumors, multifocal disease, or prior surgery altering lymphatic drainage. Additionally, the technique requires a high level of expertise and adherence to strict protocols to optimize detection rates [56].

Although SLNB is currently recommended for unifocal tumors smaller than 4 cm, ongoing research is evaluating its feasibility in patients with larger or multifocal tumors. Some studies also suggest that SLNB may serve as a reliable staging method in selected cases of the first recurrence of vulvar cancer, potentially reducing the need for complete lymphadenectomy in a broader patient population. However, further validation through prospective trials is required before expanding current guidelines. The main concern regarding SLNB in recurrence is the potential alteration of lymphatic drainage pathways due to previous surgery or radiotherapy. Prior surgery can alter lymphatic drainage, making it difficult to predict SLN locations, and fibrosis in the vulvar and groin regions may complicate the procedure. Fibrosis also increases the risk of surgical complications, such as saphenous vein injury [56,57].

At present, due to limited evidence, the safety of SLNB in vulvar squamous cell carcinoma larger than 4 cm or in multifocal tumors has not been fully established. Although recent studies have demonstrated comparable detection rates and negative predictive values to those observed in smaller, unifocal tumors, the current recommendation remains inguinofemoral lymphadenectomy [13,14,20,59].

Ongoing research aims to refine SLNB techniques further and expand their applicability. The exploration of novel tracers and imaging modalities seeks to enhance SLN detection rates and reduce false-negative occurrences. Moreover, studies are investigating the feasibility of extending SLN biopsy indications to a broader patient population while maintaining safety and efficacy. Centralizing the procedure in high-volume centers is emphasized to ensure standardized practices and optimal patient outcomes.

## 5. Conclusions

In conclusion, SLNB has become an integral component in the management of early-stage vulvar cancer, offering accurate staging with reduced morbidity compared to traditional lymphadenectomy without compromising oncological outcomes. The advancements in detection techniques and the establishment of standardized guidelines have significantly improved patient care. Future studies should focus on refining patient selection criteria, improving detection techniques, and clarifying the implications of low-volume nodal disease to further optimize outcomes for patients with vulvar cancer. Continued research and adherence to evidence-based practices are essential to further enhance the efficacy and safety of SLNB in vulvar cancer. In addition, continued quality-of-life analyses are needed to further demonstrate the short- and long-term benefits for these patients.

## Figures and Tables

**Figure 1 curroncol-32-00215-f001:**
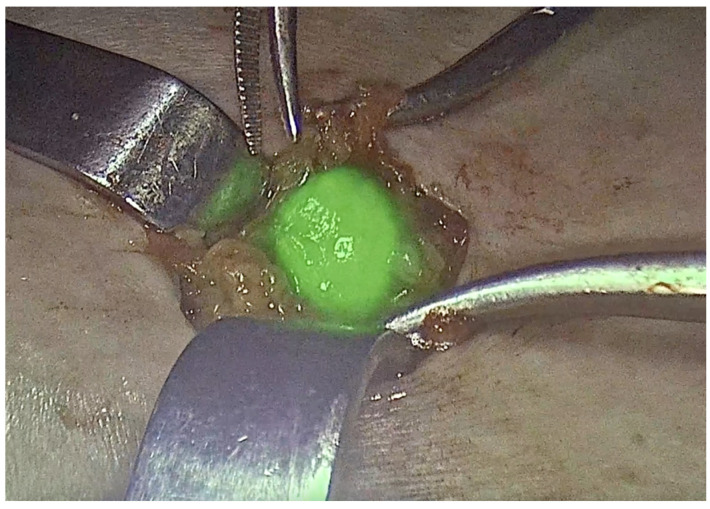
Sentinel lymph node in vulvar cancer. ICG.

**Table 1 curroncol-32-00215-t001:** Data from studies evaluating SLNB in vulvar cancer.

Author (Year)	Number of Patients	SLNB Technique	Identification Rate (%)	False Negatives (%)	Median Follow-Up (Months)	Groin Recurrences (%)	Outcome in SLN Negative Patients (%)
Van der Zee (2008) [13]	403	R + BD	97	3.3	35 (2–87)	2.3 (unifocal) 3 (including multifocal)	97 3-year DSS
Levenback (2012) [14]	452	R + BD	92.5	3.7	NA	NA	NA
Robinson (2014) [29]	69	R + BD	93	NA	58.3	4.7	NA
Grootenhuis (2016) [27]	377	R + BD	95	3.5	105 (0–179)	2.5 (unifocal)	93.5 5-year DSS 90.8 10-year DSS
Klapdor (2019) [5]	772	R or BD	94.7	5.8	33 (0–156)	4.5	82.7 3-year PFS 92.7 3-year OS

SLNB: sentinel lymph node biopsy; SLN: sentinel lymph node; R: radiotracer; BD: blue dye; DSS: disease-specific survival; PFS: progression-free survival; OS: overall survival; NA: Not aplicable.

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
