# Peer review of "Current Limitations of Sentinel Node Biopsy in Vulvar Cancer"

_curroncol, 2025, doi:10.3390/curroncol32040215_

Round 1

Reviewer 1 Report

Comments and Suggestions for Authors

In this interesting review the authors describe the state of the art regarding sentinel node biopsy (SLN) in vulvar cancer. Especially the use of the SLN in recurrent disease, multifocal disease and in tumors > 4 cm in diameter are critically reviewed. Also the sentinel node technique with the use of indocyanine green and minimal invasive surgery are discussed. Because there are no large scale prospective studies, analyzing the safety of the SLN the authors conclude that there are signs that the indications for the SLN might be extended, but prospective studies are needed to confirm this.

There are some minor flaws that must be adressed.

  1. lines 66-68: The international guidelines recommend a SLN technique in unifocal tumors < 4 cm. It is important to add the recommendation that no clinical suspicious nodes (palpation/imaging) are allowed.
  2.  Line 154; Table 1: Please check if the references in this Table are also in the reference list. For example: Robinson (2014), Klapdor (2017) are not in the list.
  3. lines 276-278 (3.3. Management of metastatic groin nodes): Please note that number of positive groin nodes is not taken into account anymore in the 2021 FIGO staging system. Please correct this.
  4. line 291-292 (3.3. Management of metastatic groin nodes). According to most guidelines, inguinofemoral nodes dissection followed by radiotherapy is only necessary in patients with > 1 macrometastasis and/or extranodal spread. Patients with one intranodal metastasis do not need adjuvant radiotherapy after a proper inguinofemoral groin dissection. Please add/correct this.
  5. lines 318-320 ( 3.3. Management of metastatic groin nodes). Please correct the english language. What is the meaning of "affectation"?
  6. lines 345-346 (3.4. Extended indications for sentinel node biopsy): Here the authors state that evidence is lacking forn the safety of the SLN procedure in recurrent vulvar cancer. This is a correct statement, but it would be better/more complete to mention that a prospective study on this subject is ongoing: (BMC Cancer. 2022 Apr 23;22(1):445. doi: 10.1186/s12885-022-09543-y)

Author Response

COMMENTS 1: lines 66-68: The international guidelines recommend a SLN technique in unifocal tumors < 4 cm. It is important to add the recommendation that no clinical suspicious nodes (palpation/imaging) are allowed.

Response 1: Thank you for pointing this out. I agree with this comment. Therefore, I/we have included it in the text (line 66-68).

COMMENTS 2: Line 154; Table 1: Please check if the references in this Table are also in the reference list. For example: Robinson (2014), Klapdor (2017) are not in the list.

Response 2: I have added Robinson 2014, reference 29 and Klapdor 2017 is not correct, the correct one is Klapdor 2019 which is reference 5; I have changed it in the table.

COMMENTS 3: line 291-292 (3.3. Management of metastatic groin nodes). According to most guidelines, inguinofemoral nodes dissection followed by radiotherapy is only necessary in patients with > 1 macrometastasis and/or extranodal spread. Patients with one intranodal metastasis do not need adjuvant radiotherapy after a proper inguinofemoral groin dissection. Please add/correct this.

Response 3: Thank you for the comment, I have change it according what you suggested in line 293-295.

COMMENTS 4: lines 318-320 ( 3.3. Management of metastatic groin nodes). Please correct the english language. What is the meaning of "affectation"?

Response 4: I have changed it by positive groin, in line 321.

COMMENTS 6: lines 345-346 (3.4. Extended indications for sentinel node biopsy): Here the authors state that evidence is lacking forn the safety of the SLN procedure in recurrent vulvar cancer. This is a correct statement, but it would be better/more complete to mention that a prospective study on this subject is ongoing: (BMC Cancer. 2022 Apr 23;22(1):445. doi: 10.1186/s12885-022-09543-y)

Response 6: Thank you for your comment, but this reference is already mentioned in line 379; reference 58.

Reviewer 2 Report

Comments and Suggestions for Authors The manuscript provides a detailed description of the application of sentinel lymph node (SLN) dissection in the treatment of vulvar cancer. The authors have reviewed a substantial amount of literature and have made a good summary of the relevant studies in recent years, especially in summarizing different research findings. To further improve the manuscript, I have the following suggestions:
  1. The title of the manuscript needs further revision. The current title is somewhat limited in scope, as this study focuses more on the application of SLN in the treatment of vulvar cancer rather than merely its limitations.
  2. Section 3.2 includes the existing SLN procedures and some staining techniques that are currently in clinical trials. The title of this section should be modified accordingly. It may be helpful to add a summarizing sentence before introducing the research on tracers in the latter half of this section.
  3. I suggest that Sections 3.3 and 3.4 be further subdivided. The authors could summarize the predictive value of SLN for lymph node metastasis and its impact on subsequent treatment choices and efficacy in two separate subsections.
  4. Some of the content in the Discussion section could be moved to Sections 3.5 and 3.6 to better organize the flow of information and avoid repetition.

Author Response

2. Point-by-point response to Comments and Suggestions for Authors

COMMENT 1: The title of the manuscript needs further revision. The current title is somewhat limited in scope, as this study focuses more on the application of SLN in the treatment of vulvar cancer rather than merely its limitations.

Response 1: Thank you for the comment, but I think the title is suitable with the content because we talk about indications of SLN in vulvar cancer but we try to focus more in SLN in recurrent vulvar cancer or tumors > 4 cm. Although the evidence in this field is more limited.

COMMENT 2:  Section 3.2 includes the existing SLN procedures and some staining techniques that are currently in clinical trials. The title of this section should be modified accordingly. It may be helpful to add a summarizing sentence before introducing the research on tracers in the latter half of this section.

Response 2: Thank you for your comment, I appreciate this, but in my opinion the title of this section is suitable with the content. Regarding the second part of the comment, I have added a summarizing sentence in line 198-199.

COMMENT 3: I suggest that Sections 3.3 and 3.4 be further subdivided. The authors could summarize the predictive value of SLN for lymph node metastasis and its impact on subsequent treatment choices and efficacy in two separate subsections.

Response 3: Thank you for your comment, I have made two separates sub-sections in section 3.3 and 3.4. Lines 273 and 323. Line 349 and 387.

COMMENT 4: Some of the content in the Discussion section could be moved to Sections 3.5 and 3.6 to better organize the flow of information and avoid repetition.

Response 4: Thank you for the comment, I have moved the last part of section 3.6 to Discussion, in line 512-517.
